# A role for Biofoundries in rapid development and validation of automated SARS-CoV-2 clinical diagnostics

Michael A. Crone [1,2,3], Miles Priestman [1,2], Marta Ciechonska [1,2], Kirsten Jensen[1,2,3], David J. Sharp[3,4], Arthi Anand[5,6], Paul Randell[7,8], Marko Storch [1,2] & Paul S. Freemont [1,2,3 ✉]

The SARS-CoV-2 pandemic has shown how a rapid rise in demand for patient and community sample testing can quickly overwhelm testing capability globally. With most diagnostic infrastructure dependent on specialized instruments, their exclusive reagent supplies quickly become bottlenecks, creating an urgent need for approaches to boost testing capacity. We address this challenge by refocusing the London Biofoundry onto the development of alternative testing pipelines. Here, we present a reagent-agnostic automated SARS-CoV-2 testing platform that can be quickly deployed and scaled. Using an in-house-generated, open-source, MS2-virus-like particle (VLP) SARS-CoV-2 standard, we validate RNA extraction and RT-qPCR workflows as well as two detection assays based on CRISPR-Cas13a and RT-loop-mediated isothermal amplification (RT-LAMP). In collaboration with an NHS diagnostic testing lab, we report the performance of the overall workflow and detection of SARS-CoV-2 in patient samples using RT-qPCR, CRISPR-Cas13a, and RT-LAMP. The validated RNA extraction and RT-qPCR platform has been installed in NHS diagnostic labs, increasing testing capacity by 1000 samples per day.

[1] London Biofoundry, Imperial College Translation and Innovation Hub, White City Campus, 80 Wood Lane, London W12 0BZ, UK. [2] Section of Structural and Synthetic Biology, Department of Infectious Disease, Imperial College London, London SW7 2AZ, UK. [3] UK Dementia Research Institute Centre for Care Research and Technology, Imperial College London, London, UK. [4] Department of Brain Sciences, Imperial College London, Hammersmith Hospital, Du Cane Road, London W12 0NN, UK. [5] Histocompatibility and Immunogenetics Laboratories, Department of Infection and Immunity, North West London Pathology, London, UK. [6] Imperial College Healthcare NHS Trust, Hammersmith Hospital, Du Cane Road, London W12 0HS, UK. [7] Department of Infection and Immunity, North West London Pathology, London, UK. [8] Imperial College Healthcare NHS Trust, Charing Cross Hospital, Fulham Palace Road, Hammersmith, London W6 8RF, UK. ✉email: p.freemont@imperial.ac.uk

Following the report of a case in Wuhan on 31 December 2019, the rapid spread and highly infectious nature of the newly emerged coronavirus has resulted in a worldwide pandemic, as declared by the World Health Organization (WHO) on 11 March 2020[1]. The causative agent of Coronavirus Disease 2019 (COVID-19) has been classified as severe acute respiratory syndrome coronavirus 2 (SARS-CoV-2) and is closely related to the severe acute respiratory syndrome (SARS) and Middle East respiratory syndrome (MERS) coronaviruses, which were responsible for outbreaks in 2003 and 2012, respectively[2]. As of 8 July, there have been 11,850,000 SARS-CoV-2-confirmed cases worldwide, with 544,000 deaths in 213 countries and territories[3]. The fast rate of SARS-CoV-2 human-to-human transmission has resulted in an unprecedented need for diagnostic testing, placing a great strain on public health departments in every country. Diagnostic testing is essential not only for the identification of infection in patients but also for tracking and containment of viral spread within communities, testing of unresolved cases, and daily screening of medical frontline workers.

Automated workflows are highly preferable over manual protocols to achieve meaningful throughput, diagnostic precision, and to exclude human error from the sample processing pipeline. Typical automated systems such as the Roche cobas® unit can process hundreds of samples per day with minimal staff support, while ensuring uniform processing and sample tracking. As with other similar automated diagnostic testing platforms, they are costly, not available in the numbers needed to process hundreds of thousands of samples per day in the United Kingdom and currently suffer from reagent supply shortages. Thus, an urgent need has arisen for the adaptation of alternative automated liquid-handling platforms and diagnostic test approaches and workflows, ideally designed in an open and modular way to allow for diversification of reagent supply away from mainstream and overstretched reagent sources.

Many research institutions around the world have established non-commercial Biofoundries, which offer integrated infrastructure including state-of-the-art automated high-throughput (HT) equipment to enable the design-build-test cycle for large-scale experimental designs in synthetic biology[4]. This infrastructure, in combination with technical expertise in molecular biology, analytics, automation, engineering, and software development, provides an excellent, self-sufficient, and agile capability to quickly establish platforms for prototyping biological testing standards and developing liquid-handling workflows, such as those needed for automated diagnostic testing of SARS-CoV-2. In the London Biofoundry, we rapidly re-configured existing liquid-handling infrastructure to establish an automated HT SARS-CoV-2 diagnostic workflow with reverse-transcriptase quantitative PCR (RT-qPCR), CRISPR-Cas13a, and RT-loop-mediated isothermal amplification (RT-LAMP)-mediated outputs.

Armoured RNA particles are non-infectious RNA virus surrogates consisting of MS2 bacteriophage capsids containing an RNA template of choice[5]. Previously, they have been employed as diagnostic reference tools for the detection of respiratory viruses such as Influenza A and B, as well as SARS-CoV[6,7]. The particles can be handled in Biosafety Level 1 laboratories and thus do not require specialist equipment as is the case for SARS-CoV-2 or live patient samples. Furthermore, they are nuclease resistant, having been shown to be highly stable in plasma, nasopharyngeal secretions, faeces, and water, and simulate the presence of a real viral target[7]. Here we engineer and characterize such synthetic virus-like particles (VLPs) as a standard simulating SARS-CoV-2. The particles contain the genomic RNA segment encoding the full-length N protein for validation using the N1, N2, and N3 primer–probe sets specified by the Centers for Disease Control

and Prevention (CDC) 2019-Novel Coronavirus (2019-nCoV) real-time RT-PCR Diagnostic Panel[8,9].

We use the SARS-CoV-2 VLPs as a quantitative standard and processing control to design and optimize an automated nucleic acid test (NAT) diagnostic workflow encompassing viral RNA extraction and one-step RT-qPCR. The optimized workflow has been validated using patient samples and results show high correlation with accredited diagnostic laboratory test results. Next, we modularize the workflow to anticipate deficits in the reagent supply chain and availability of qPCR equipment. To this end, we implement multiple off-the-shelf RNA extraction kits and assessed the quality of several RT-qPCR reagent suppliers. Finally, we develop automated workflows for CRISPR-Cas13a diagnostic[10] and colorimetric isothermal amplification (RT-LAMP) systems[11], as alternative SARS-CoV-2 detection methods that may aid in the expansion of current diagnostic capacity to population testing.

On 7 July 2020, the United Kingdom processed 40,321 samples for diagnostic testing in the National Health Service (NHS), which has increased from 10,412 on 1 April when reporting started and from 13,097 when both of our platforms were in use for frontline testing on 28 April[12]. By automating the SARS-CoV-2 NAT workflow, we report an average sample processing rate of ~1000 samples per platform per day, which can be easily modified and scaled to 4000 samples per day. The complete platform is rapidly deployable and its footprint requires a small laboratory bench, thus making it easily portable and suitable for testing in areas of low population density. With our workflow validated and implemented in NHS diagnostic laboratories, our work has helped increase the current testing capacity in London and will provide a blueprint and validation for Biofoundries and interested laboratories globally.

## Results

**VLP preparation and characterization**. Recombinant MS2 bacteriophage VLPs carrying the SARS-CoV-2 N-gene were produced in *Escherichia coli* from an expression plasmid using protocols described previously and modified to transcribe and package the RNA for the SARS-CoV-2 N protein (Fig. 1a)[13,14]. The assembled MS2-SARS-CoV-2 VLPs were purified and treated with DNase and RNase, to ensure the preparation was free from template DNA and RNA contamination. The purity of the sample was analysed by SDS-polyacrylamide gel electrophoresis (Fig. 1b). We determined the VLP size distribution using dynamic light scattering (DLS) to be ~27 nm (Fig. 1c), which matched well with a previously characterized MS2 VLP construct[13]. Next, we employed reverse-transcriptase droplet digital PCR (RT-ddPCR) to obtain absolute quantities of three serial dilutions of VLPs. Heat lysis has been shown to be effective in releasing RNA from armoured RNA VLPs at levels comparable to commercially available extraction kits[7]. RNA encoding the CoV-2 N protein was extracted from the VLPs at 95 °C for 5 min, followed by amplification using the CDC N1 primer–probe set. The released RNA was serially diluted and absolute quantification was performed. The RT-ddPCR quantification method involves partitioning of the sample into thousands of droplets, which individually contain single amplification reactions using the N1 probe. VLP concentration can then be derived using Poisson distribution statistics to determine absolute particle concentration in each dilution. This method allows for highly accurate and precise sample quantification without the need for a standard curve. We analysed the purified MS2-SARS-CoV-2 VLP absolute concentration in serial tenfold dilutions, which were found to contain 250, 25, and 2.5 copies/µl, respectively (Fig. 1d).

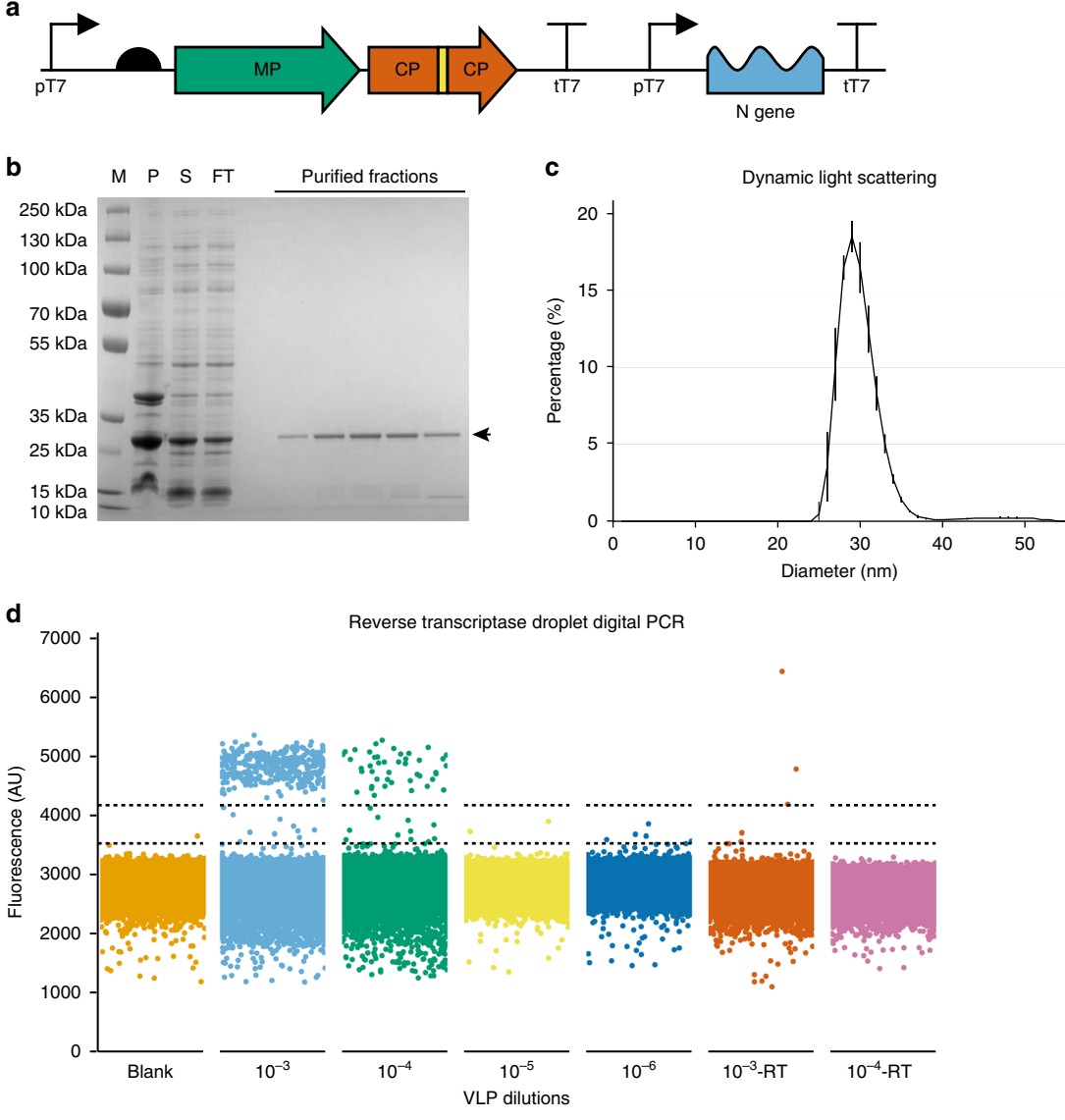

**Fig. 1 MS2-SARS-CoV-2 VLP production and characterization. a** Schematic of the genetic construct of the engineered MS2-SARS-CoV-2 N-gene VLP encompassing the MS2 maturation protein (MP) in green and coat proteins (CP) in orange, linked via His-tag (yellow), under transcriptional control of the T7 promoter and T7 terminator sequences. The SARS-CoV-2 N protein RNA is packaged using a downstream *pac site*. Schematic created using DNAplotlib[31]. **b** The VLP constructs were expressed in *E. coli* and purified using HiTrap® TALON® Crude and HiTrap® Heparin columns. SDS-PAGE analysis of the purification steps includes a protein marker (M) followed by pellet (P) and soluble fraction (S) of the cell lysate, followed by the column flow through (FT) and protein elution fractions, where the CP-His-CP dimer (~28 kDa) is indicated by an arrow. **c** Purified VLPs were analysed by dynamic light scattering (DLS), which showed a uniform particle population of ~27 nm. Error bars represent the SD of three technical replicates. **d** Reverse-transcriptase droplet digital PCR (RT-ddPCR) was performed for absolute quantification of the purified VLPs. Serial dilutions of 1, 10, and 100 thousand-fold of the purified VLPs in the presence and absence of a reverse transcription (RT) enzyme were analysed. Droplets were clustered using a threshold determined using a python implementation of an online tool (http://definetherain.org.uk). Dotted lines represent the cut offs for the positive and negative clusters. Any data points between the two dotted lines are considered droplet "rain". Source data are available in the Source Data file.

**VLP validation as a SARS-CoV-2 standard by RT-qPCR.** One-step RT-qPCR is currently the gold standard for detection of nucleic acids in molecular diagnostic tests due to its sensitivity, robustness, dynamic range, HT capability, and affordability. It is the current method of choice for the detection of SARS-CoV-2 in the UK and around the world. To demonstrate the utility of MS2-SARS-CoV-2 VLPs as a standard for optimizing and validating automated NAT diagnostic workflows, we assessed whether they could be reliably detected via RT-qPCR using the CDC 2019-nCoV Diagnostic Panel primer–probe set. This primer–probe set was used for all RT-qPCR validation experiments and the setup of the RT-qPCR reactions was automated. We extracted the SARS-

CoV-2 N protein RNA encapsulated by the MS2 VLP using heat lysis from the serial dilutions quantified via RT-ddPCR and performed One-Step RT-qPCR using the TaqPath master mix (Thermo Fisher Scientific). Three biological replicates were completed to assess the robustness of using the MS2-SARS-CoV-2 VLPs as control RNA (Fig. 2a). The quantified VLP dilutions were also used to generate a standard curve to aid in assessment of viral RNA purification efficiency and to estimate the limit of detection (LoD) of our automated workflow (Fig. 2b).

The current unprecedented demand for the one-step RT-qPCR master mix to detect SARS-CoV-2 may result in disruption of the laboratory reagent supply chain. Here we used our MS2-SARS-

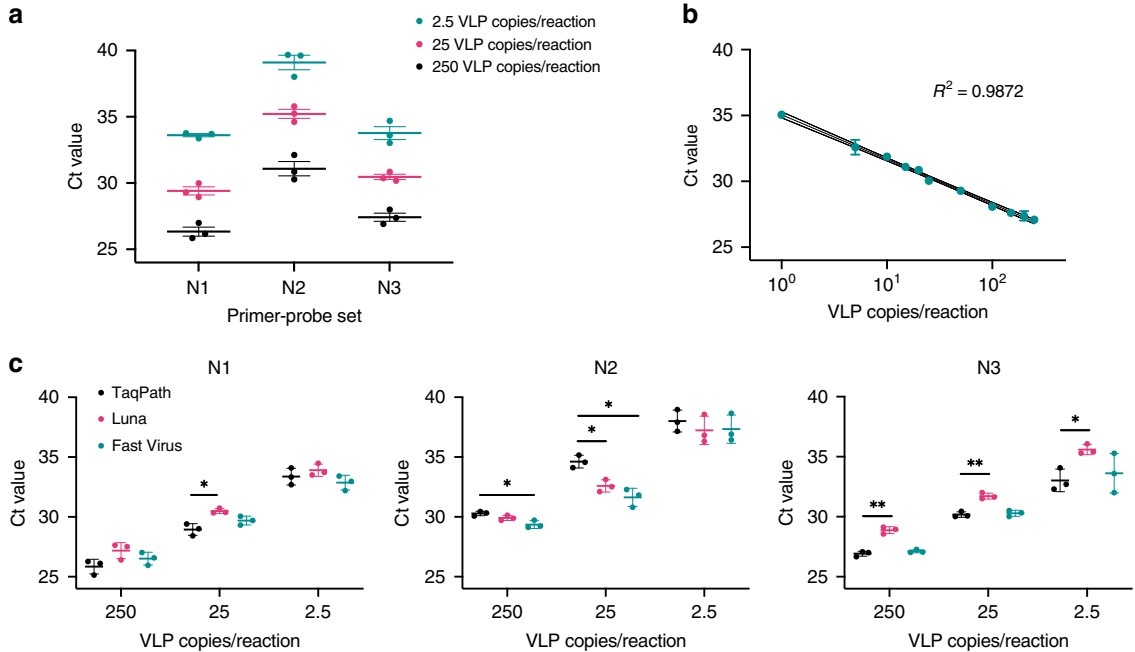

**Fig. 2 MS2-SARS-CoV-2 VLP detection with multiple target primer–probe sets and qPCR master mixes. a** VLP dilutions of 2.5, 25, and 250 copies per reaction were analysed by one-step RT-qPCR using the CDC primer–probe sets N1, N2, and N3 with the TaqPath master mix (Thermo Fisher Scientific) and reported as Ct values. **b** A Ct value standard curve for VLP concentrations of 1, 5, 10, 15, 20, 25, 50, 100, 150, 200, and 250 VLP copies per reaction was determined using the N1 primer–probe set and the TaqPath master mix. **c** VLP dilutions of 250, 25, and 2.5 copies per reaction were analysed using the TaqPath, Luna Universal (NEB), and Fast Virus (Thermo Fisher Scientific) RT-qPCR master mixes with the N1, N2, and N3 CDC primer–probe sets. All measurements in **a** and **c** are reported as mean ± SE of $n = 3$ independent experiments with three technical replicates. Measurements in **b** are reported as mean ± SD and are representative of $n = 2$ independent experiments with three technical replicates. Statistical difference between the TaqPath and Luna, as well as TaqPath and Fast Virus master mix Ct values was analysed using an unpaired two-sided $t$-test with (black star) indicating $p < 0.05$ and (double black star) $p < 0.01$. Source data are available in the Source Data file.

CoV-2 standards to demonstrate modularity of the one-step RT-qPCR detection method by comparing detection reproducibility between three commercially available master mixes using the CDC Diagnostic Panel N1, N2, and N3 primer–probe sets. We report that Ct values achieved using the Virus Fast One-Step (Thermo Fisher Scientific) master mix closely match those generated with the gold standard TaqPath master mix from the same supplier (Fig. 2c). Ct values obtained using the Luna Universal RT-qPCR kit supplied by New England Biolabs differ slightly from the TaqPath master mix. Although all three primer–probe sets achieved similar Ct results, N1 produced the lowest Ct values with least variability, as previously reported by Vogels et al.[15]. The N2 primer–probe set produced higher Ct values and exhibited more variability between replicates for all three RT-qPCR master mix options. Based on its higher sensitivity, we chose the N1 primer–probe set for validation of our RNA extraction and virus detection workflows.

**CRISPR-Cas13a as an alternative to one-step RT-qPCR.** To further expand the modular nature of the automated platform, we assayed an alternative to the standard RT-qPCR detection method by employing a CRISPR-Cas13a NAT. This approach, based on the specific high-sensitivity enzymatic reporter unlocking (SHERLOCK) method, was designed to identify and amplify target sequences of the CoV-2 N-gene RNA packaged within the MS2-SARS-CoV-2 VLP[16]. Briefly, similar to RT-qPCR, the initial step of this method relies on the reverse transcription and amplification of the target RNA. Here we employed the above-mentioned CDC diagnostic primer sets N1, N2, and N3 (with forward primers 5′ extended with a T7 promoter sequence) together with a one-step RT enzyme mix to generate cDNA from

N-gene RNA released from the VLPs. However, unlike qPCR, the subsequent step includes transcription of the amplified segment to RNA and incubation with a CRISPR RNA (crRNA), in this case complementary to the N region of CoV-2, together with a purified recombinant Cas13a protein derived from *Leptotrichia wadei* (LwCas13a). Upon recognition and binding of the target sequence by crRNA, Cas13a exhibits RNase activity not only for this complex but also collateral activity for any RNA in its vicinity. Taking advantage of this nonspecificity, a quenched fluorescent probe can be added and subsequently cleaved by activated Cas13a, thus generating a quantitative signal that can be detected using a standard fluorescence microplate reader. By applying this technology, we assayed MS2-SARS-CoV-2 RNA released through heat lysis and were able to detect the CoV-2 RNA sequence at 250, 25, and 2.5 VLP copies per reaction (Fig. 3). This low LoD was comparable to all of the one-step RT-qPCR master mixes reported above, although viral load quantification is difficult. Thus, we propose this method may be used to substitute current RT-qPCR diagnostic workflows, as it does not require qPCR equipment and is highly amenable to HT automated workflows. Furthermore, it enables 10- to 100-fold increased throughput when performed in high-density assay plates, with accurate diagnostic test readout available in just a few minutes using a standard fluorescence microplate reader. Another advantage of our CRISPR workflow is the possibility of identifying specific viral serotypes in a multiplexed strain-specific diagnostic, which would provide additional information for clinical management.

**Colorimetric RT-LAMP as an alternative to one-step RT-qPCR.** As an additional alternative detection methodology, we adapted the previously described colorimetric RT-LAMP assay[11]

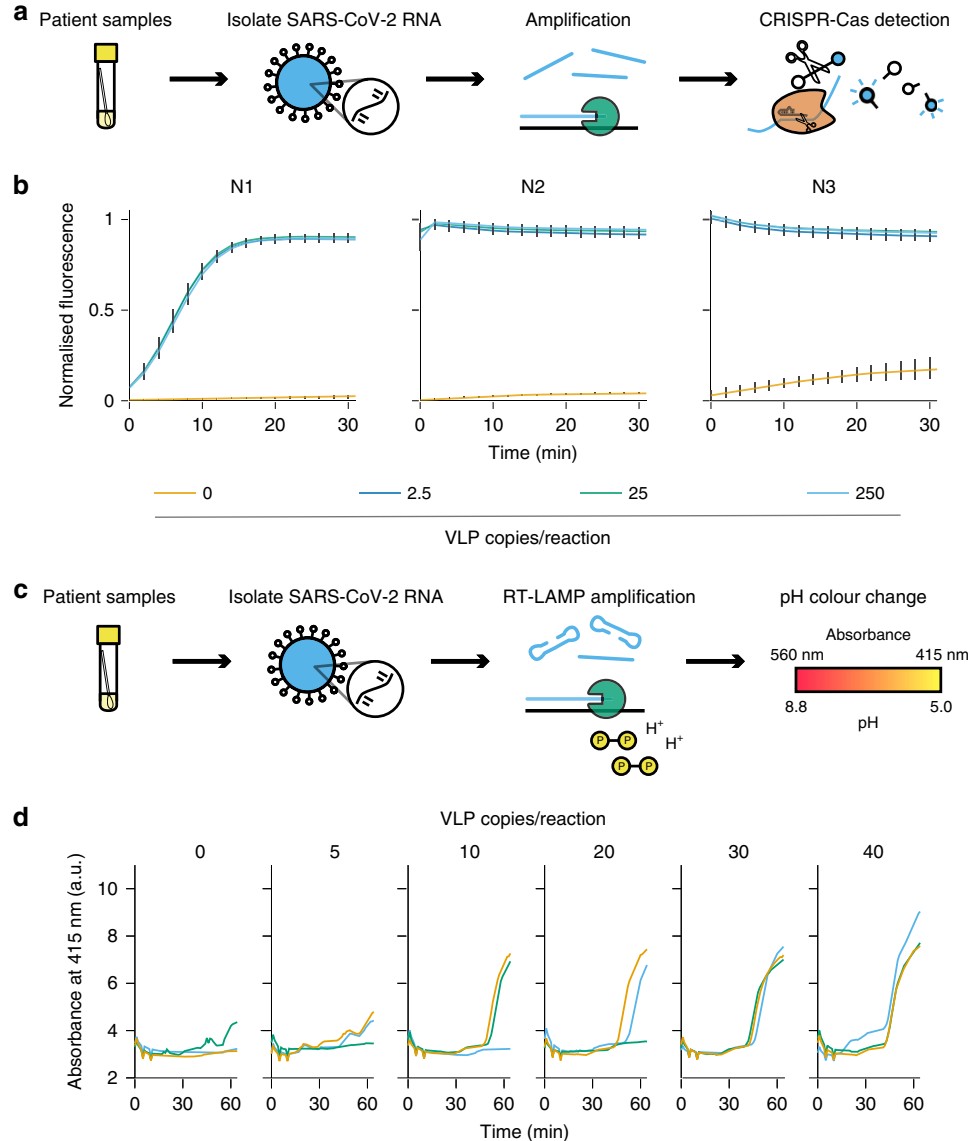

**Fig. 3 SARS-CoV-2 N-gene target RNA detection by CRISPR-Cas13a and RT-LAMP diagnostic systems. a** Schematic of the CRISPR-Cas13a nucleic acid detection workflow from patient samples. Viral RNA is amplified using target-specific primers and detected with target-specific crRNA activating Cas13a to collaterally cleave a fluorescent probe. **b** CDC N1, N2, and N3 primer sets were employed to amplify the N-gene RNA released from MS2-SARS-CoV-2 VLPs at 2.5, 25, and 250 copies per reaction. The CRISPR-Cas13a detection time course was analysed using a fluorescence microplate reader. Error bars represent the SEM of $n = 3$ biologically independent amplification reactions and four CRISPR detection technical replicates. **c** Schematic of the RT-loop-mediated isothermal amplification (RT-LAMP) diagnostic workflow using target-specific LAMP primers. The isothermal amplification of a target results in the acidification of the RT-LAMP master mix and a subsequent pH-associated colour change that is detected using a microplate reader. **d** Time-course detection of three replicate RT-LAMP reactions using the MS2-SARS-CoV-2 VLPs at 0, 5, 10, 20, 30, and 40 copies per reaction, performed at 65 °C using the BMG CLARIOstar Plus microplate reader. Source data are available in the Source Data file.

for use with HT automation. In this approach, a pyrophosphate moiety and hydrogen ion are produced for every nucleotide that is incorporated into the PCR product during each amplification step. The release of hydrogen ions results in a pH change in a minimally buffered reaction which can be visually determined using dyes such as phenol red[17] and by measuring absorbance (Fig. 3c). Here we employed the Labcyte Echo platform to set up RT-LAMP reactions using various concentrations of VLP N-gene RNA template in 384-well microplates. These were incubated at 65 °C in a microplate reader and absorbance at 415 nm was measured over time. We demonstrated that the presence of the target sequence can be detected reliably down to at least 30 copies per reaction, although viral load quantification using RT-LAMP is not currently possible. The use of automation coupled with the

speed and affordability of the RT-LAMP workflow provides an excellent alternative to qPCR diagnostic NATs in a format that is highly amenable to ultra-HT workflows (Fig. 3d).

**Automated workflow development and validation.** Automation of clinical laboratory diagnostics has been essential to increase sample processing throughput, minimize run times, standardize sample processing, maximize accuracy and reproducibility, and to reduce human error. The unprecedented need for diagnostic testing imposed by the SARS-CoV-2 pandemic has resulted in a bottleneck in sample processing throughput. To increase patient sample turnaround time, we have developed an automated diagnostic workflow including RNA extraction and RT-qPCR

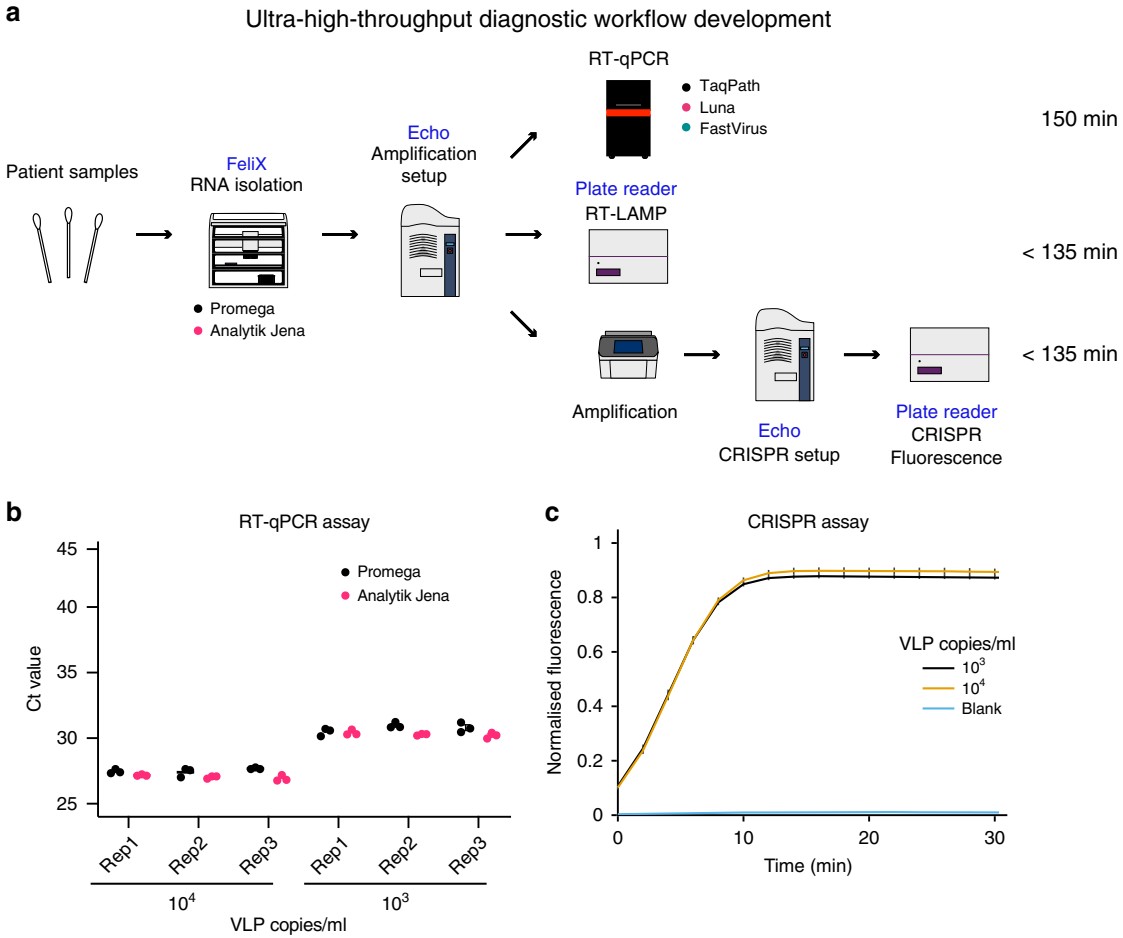

**Fig. 4 SARS-CoV-2 platform development using the MS2-SARS-CoV-2 VLP standard. a** Schematic of the modular platform for the detection of SARS-CoV-2. Viral RNA is isolated using the Analytik Jena FeliX liquid handler with the Analytik Jena or Promega RNA extraction kits. Sample RNA is transferred and detection reactions are set up using the Labcyte Echo platform. These include qPCR, validated for the TaqPath (Thermo Fisher Scientific), Luna (NEB), and Fast Virus (Thermo Fisher Scientific) RT-qPCR master mix options, RT-LAMP nucleic acid detection, and the CRISPR-Cas13a diagnostic workflow. **b** Automated RNA extraction was developed using VLP dilutions of $10^3$ and $10^4$ copies/mL for the Analytik Jena and Promega RNA extraction kits. Efficiency of the extractions using both kits was analysed by RT-qPCR with the CDC N1 primer–probe set using the TaqPath master mix. **c** Dilutions of VLPs used for RT-qPCR in **b** were analysed using the CRISPR NAT to demonstrate the use of this workflow as an alternative diagnostic option. Error bars in **b** represent mean ± SE of $n = 3$ biologically independent samples with three technical replicates. Error bars in **c** represent mean ± SE of $n = 3$ independent amplification replicates and four technical replicates for CRISPR detection. Source data are available in the Source Data file.

using elements of the full synthetic biology technology stack available at the London Biofoundry (Fig. 4a). We employed the MS2-SARS-CoV-2 VLPs as a process control to optimize and validate our automated clinical diagnostic workflow encompassing RNA extraction and the RT-qPCR and CRISPR Cas13 detection methods. To design full-factorial experiments, track randomized samples, and to document optimization and validation experiments, we used the Riffyn platform in combination with SAS JMP software. VLP RNA extraction was optimized for the Analytik Jena CyBio FeliX liquid-handling platform for the standard 96-well plate format using the innuPREP Virus DNA/RNA Kit-FX. We prepared two VLP dilutions of 1000 copies/mL and 10,000 copies/mL, to simulate viral load amounts found in patient samples[18]. Using the optimized FeliX extraction protocol, we were able to isolate RNA from the test VLP dilutions within 60 min. The automated workflow takes advantage of magnetic bead-based nucleic acid extraction and eliminates laborious and time-consuming column-based binding and spinning steps. Although we tested three biological replicates of the two dilutions, this workflow is designed for the concurrent processing of 96 samples. We project that employing the FeliX RNA extraction

protocol in a 96-well format and using one liquid-handling device can result in the processing of 1000 samples in 12 h, including extra time for reagent and extraction kit refilling and patient sample plate loading. This workflow requires minimal user intervention and is therefore highly scalable.

Next, we employed RT-qPCR to determine RNA extraction efficiency. To this end, we used a combination of Riffyn and SAS JMP software to generate and track randomized pick lists for the Labcyte Echo 525 acoustic liquid-handling platform, which was used to automate the TaqPath master mix, primer–probe, and sample transfer into 96-well qPCR plates. Plate-handling time from RNA extraction to qPCR launch was ~10 min when using the Echo 525 followed by a qPCR running time of ~70 min. The extracted VLP RNA assayed by RT-qPCR resulted in Ct values that were in agreement with those achieved with VLP heat lysis described above for the same concentrations, suggesting a high efficiency of extraction using the automated platform (Fig. 4b). Total running time from beginning of the RNA extraction to obtaining RT-qPCR results was ~2 h and 45 min.

A key advantage of the Analytik Jena CyBio FeliX liquid-handling platform over other dedicated diagnostic platforms is its

programmability. This allows a reagent-agnostic approach to be developed, permitting robust supply chains to be established. To this end, we also implemented and optimized RNA extraction using an alternative kit—Maxwell HT Viral TNA (Promega)— using the same platform. This required minor adjustments to match manufacturer recommendations for optimal volumes and empirically determined mixing steps. Crucially, the hardware is identical, the plasticware is identical, and the output is identical thus requiring no changes to working practice. The same VLP dilutions were used to allow comparison of RNA extraction efficiency between the innuPREP Virus DNA/RNA and the Maxwell HT Viral TNA kit by RT-qPCR (Fig. 4b). The Ct values achieved from RNA extracted using the Promega kit were broadly 0.5 cycles higher than those for the Analytik Jena kit ($27.52 \pm 0.05$ and $27.04 \pm 0.04$, respectively, for the high VLP concentration, $p = 2.90 \times 10^{-7}$; and $30.76 \pm 0.08$ and $30.30 \pm 0.05$ for the lower concentration, $p = 2.02 \times 10^{-4}$; ±SEM, paired $t$-test).

In parallel, a CRISPR-Cas13a workflow was tested for situations where the number of qPCR machines, but not PCR machines, may be a limiting factor. Samples were pre-amplified with one-step RT-PCR master mix and 0.25 μL of each PCR product was added to 4.75 μL of Cas13a crRNA master mix using the Labcyte Echo 550. Reactions reached saturation in positive samples within ~10 min (Fig. 4c). Although this approach slightly lengthens the approximate running time from RNA extraction to obtaining a diagnostic test result to ~3 h (reduced to ~2 hr and 15 min when using isothermal amplification techniques), it provides an alternative detection methodology that is more easily scaled than qPCR workflows.

**Validation of the automated platform with patient samples.** After demonstrating that the workflow could detect VLPs loaded with SARS-CoV-2 RNA at clinically relevant concentrations, we validated the platform with 173 patient samples obtained from North West London Pathology (NWLP). We compared our extraction (Analytik Jena innuPREP Virus DNA/RNA Kit) and qPCR workflow to that of NWLP at the time (a multiplexed-tandem PCR workflow). Patient samples were stored at room temperature for no more than 48 h after the initial analysis by NWLP before they were purified on our platform. We assayed 5 μL of purified RNA from each patient sample (Fig. 5a) and showed good correlation ($R^2 = 0.8310$) between our results and the test used by NWLP (Fig. 5b). Of 173 samples tested, we were able to match 49 positive and 120 negative samples, with 3 samples detected by NWLP only and 1 sample detected using our workflow only. Notably, these four samples showing a lack of concordance were all close to the LoD.

We then compared the Promega Maxwell HT Viral TNA extraction kit to the previously validated Analytik Jena innuPREP Virus DNA/RNA Kit workflow with a second set of 65 patient samples. We observed high correlation ($R^2 = 0.9357$) between Ct values for the same samples when processed with either the Promega or Analytik Jena extraction kits (Fig. 5c). This highlights the strength of the platform providing consistent results for diverse reagent kits and supply chains. The more reagent kits are validated, the more resilience can be added through redundancy.

Finally, previously described detection assays[11,19] were demonstrated in a HT-compatible format. Samples were extracted using the FeliX liquid-handling protocol and resulting elution samples were transferred to plates certified for acoustic liquid handling. Miniaturized reactions were then set up to enable alternative detection modalities that are HT-compatible. The CRISPR-Cas13a workflow with Reverse Transcription Recombinase Polymerase Amplification (RT-RPA) was shown to give a semi-quantitative output down to around 200 copies per reaction

(Fig. 5d, Supplementary Fig. 1a, and Supplementary Table 1) with one amplification replicate (4 of 12 CRISPR reactions) stochastically showing detection below this limit (Supplementary Fig. 1a). In addition, colorimetric RT-LAMP was shown to be more sensitive than RPA-CRISPR with at least two replicates showing detection for as low as ~26 copies per reaction (Fig. 5e, Supplementary Fig. 1b, and Supplementary Table 1).

## Discussion

In this study, we have been able to quickly repurpose automated liquid-handling infrastructure in the London Biofoundry to establish two frontline SARS-CoV-2 testing platforms, which are now operational in two London hospitals with a testing capacity of 2000 samples per day. We have also developed CRISPR and colorimetric LAMP-based workflows and have established a SARS-CoV-2 VLP standard, which has allowed us to validate the workflows within our biofoundry before implementation. During this process, we have identified a number of opportunities where biofoundries can be very effective in quickly providing increased SARS-CoV-2 testing capacity.

One major issue with standard diagnostic laboratory workflows is an over-reliance on a small number of manufacturers for infrastructure. For example, integrated platforms that allow HT sample processing—including automated patient sample nucleic acid extraction—are available from a few manufacturers, such as Roche, Abbott, Hamilton, Thermo Fisher, and Qiagen. However, at a time of unprecedented sample processing need, such as that imposed by the global COVID-19 pandemic, innovative approaches and non-traditional entities such as biofoundries, academic labs, start-ups, and small and medium-sized enterprises can greatly expand testing options to add not only increased capacity, but also improved supply chain resilience[20,21]. Biofoundries are agile facilities with a highly skilled workforce and cutting-edge equipment that can rapidly respond to such new challenges. Typically, they are not-for-profit institutions and therefore can evaluate different strategies unconstrained by commercial considerations. Their aim is to develop and apply purpose-built laboratory automation platforms, with an emphasis on versatile equipment, which can be adapted to a variety of synthetic biology workflows and support the translation of the latest scientific developments at their hosting research institutes and beyond. As lessons from management of the outbreak in Wuhan are beginning to emerge, it is becoming clear that automated diagnostic workflows, such as those implemented at the Huo-Yan diagnostic laboratory for processing 10,000 tests per day, play a pivotal role in containment of the virus[22].

When developing diagnostics for high-pressure pandemic scenarios, it is critical to create workflows that are modular and offer multiple contingency options, as reagent supply can quickly become a limiting factor to sample processing. Here we describe the rapid development of a HT diagnostic platform for the detection of SARS-CoV-2, using a synthetic VLP developed in-house under biosafety level 1 conditions. We use a versatile automated liquid-handling device, the Analytik Jena CyBio FeliX, and validate two RNA extraction kits, multiple qPCR master mixes, as well as CRISPR- and colorimetric LAMP-based workflows, all in under 4 weeks. Importantly, we also validate the RNA extraction and RT-qPCR assay using patient samples, demonstrating a good correlation between a currently used clinical laboratory test for SARS-CoV-2 and our modular workflow. The framework provided for the validated platform may be further extended by alternative extraction and detection methodologies as well as in-house production and optimization of kit components[23,24]. This toolkit increases the resilience of the SARS-CoV-2 NAT in case of shortages in extraction materials, RT-qPCR

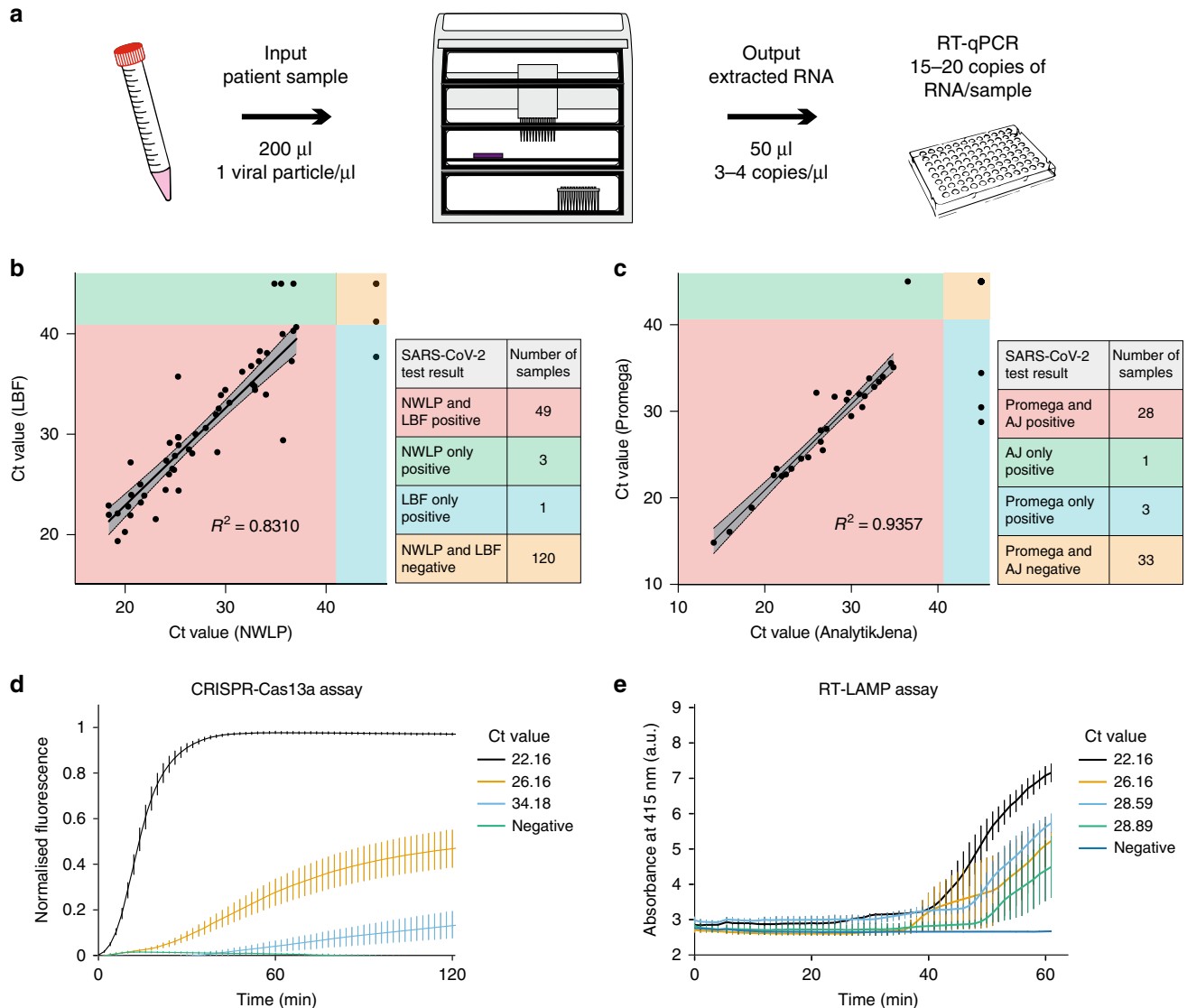

**Fig. 5 SARS-CoV-2 platform validation using patient samples. a** Schematic of a typical workflow tracking viral copy number from a hypothetical patient sample where an input of 200 μL with a minimum 1 particle/μL results in 3–4 copies of extracted RNA per microliter resulting in a range of 15–20 RNA copies per qPCR reaction. **b** Comparison of RT-qPCR Ct value results for the same 173 patient samples obtained by the North West London Pathology (NWLP) laboratory and the London Biofoundry (LBF), based on the VLP workflow using the Analytik Jena RNA extraction kit and CDC N1 primer–probe set. **c** Validation of the Promega RNA extraction kit using 65 patient samples with the Analytik Jena RNA extraction workflow previously validated in **b**. Error bands in **b** and **c** represent the 95% confidence interval. **d** Validation of the CRISPR NAT using patient samples with Ct values determined via RT-qPCR. Error bars represent the mean ± SE of $n = 3$ independent amplification replicates and four technical replicates for CRISPR detection. **e** Validation of the RT-LAMP colorimetric NAT using patient samples with Ct values determined via RT-qPCR. Error bars represent the mean ± SE of $n = 3$ independent amplification reactions. Source data are available in the Source Data file.

master mix, and laboratory equipment availability. This modularity is created not only by generating custom protocols for several commercially available kits, but also by adapting the CRISPR-Cas13a detection and colorimetric LAMP systems to HT SARS-CoV-2 diagnostic testing. CRISPR-based detection technologies are also currently being developed by Sherlock Biosciences and Mammoth Biosciences to provide at-home point-of-care testing kits[10,19,25], as well as in CARMEN-Cas13, a microwell array that multiplexes virus detection[26].

Our workflow is easy to scale up, cost-effective, and can provide similar output capacity to that offered by the gold standard of commercial automated systems. For example, a single FeliX liquid handler and qPCR thermocycler can match the largest state-of-the-art Roche cobas® 8800 platform, which can process 960 samples in eight hours. In addition, excess viral RNA

remaining from the FeliX patient sample RNA extraction can be diverted to alternative analysis workflows such as next-generation sequencing, which is not possible for some commercial platforms. Finally, our automated RNA extraction and qPCR workflow requires minimal specialist training and can be launched within one day. It is currently installed—and used—in NHS diagnostic labs, where patient sample testing has been validated against large commercially available platforms, matching their precision and throughput.

Although NHS labs currently rely on qPCR workflows for all SARS-CoV-2 diagnostic testing, the potential of alternative detection technologies would allow for HT testing for population screening and in low-resource settings. Miniaturizing LAMP and CRISPR reactions results in a slight loss of sensitivity and therefore may not be suitable for making diagnostic decisions

where qPCR capacity is available; however, their isothermal incubation allows for thousands of samples to be tested simultaneously (Comparison of Methodologies in Supplementary Table 2). LAMP is a particularly attractive technique, because it has also been shown to be sensitive with heat-inactivated samples, removing the bottleneck of RNA extraction[27]. Furthermore, these solutions can be deployed for community testing in low-resource settings or at the point-of-care without expensive equipment requirements.

Engineered VLPs have been widely reported and commercially used as controls and standards in nucleic acid-based diagnostic tests (Asuragen), and have been developed as antigen epitopes in serological assays, where they are used to detect patient antibodies (Native Antigen)[28,29]. MS2 VLPs carrying RNA payload, such as those used in this study for the detection of SARS-CoV-2 N-gene RNA, provide a quick and reproducible system for generating extremely stable NAT controls. As such, we have purified and quantified large batches that are ready to be shared with and employed by others for diagnostic test development that relies on viral RNA detection. Furthermore, our VLP production and characterization workflow can be modified to rapidly generate new controls mimicking emerging viral threats, thus enhancing preparedness for the development of new diagnostics in future epidemic or pandemic scenarios. In addition, automation equipment available in biofoundries can be used for large-scale testing of antigen-presenting VLPs in developing antibody-based enzyme-linked immunosorbent assay diagnostics and for performing HT antiviral drug screens. The London Biofoundry is a founding member of the Global Biofoundry Alliance, which currently encompasses 26 such entities worldwide[30]. This network allows for easy sharing of reagents, protocols, and technical know-how. Therefore, automated diagnostic workflows developed by one partner can be quickly replicated around the world and increase capacity for testing and drug development to help counteract and prevent the global spread of emerging pathogens.

## Methods

**Primers and probes**. Primers and probes were ordered from IDT or Biolegio and can be found in the Supplementary Information in Supplementary Tables 3, 4, and 5.

**VLP preparation**. The nucleic acid sequence of the N-gene of SARS-CoV-2 (accession number: NC_045512) was ordered from GeneArt (Thermo Fisher Scientific). The N-gene was cloned into a MS2 VLP expression plasmid backbone (Addgene #128233) using Type IIs assembly. The sequence-verified (Eurofins Genomics) plasmid (Addgene #155039) was then transformed into Rosetta 2 (DE3) pLysS cells (Merck Millipore). An overnight culture was used to inoculated 200 mL of Terrific Broth (Merck) supplemented with 35 μg/mL of Chloramphenicol (Merck) and 50 μg/mL of Kanamycin (Merck), and grown at 37 °C, 200 r.p.m. until an OD of 0.8. The culture was induced by supplementing with 0.5 mM IPTG (Merck) and grown at 30 °C for a further 16 h. Cells were collected at 3220 × g at 4 °C and stored at −20 °C for later purification.

All protein purification steps were performed at 4 °C. The cell pellet was resuspended in 4 mL Sonication Buffer (50 mM Tris-HCl pH 8.0, 5 mM MgCl$_2$, 5 mM CaCl$_2$, and 100 mM NaCl) with 700 U RNase A (Qiagen), 2500 U BaseMuncher (Expedeon), and 200 U Turbo DNase (Thermo Fisher Scientific). The cells were sonicated for a total of 2 min (50% amplitude, 30 s on, 30 s off) on wet ice. The lysate was then incubated for 3 h at 37 °C. The lysate was centrifuged at 10,000 × g for 10 min at room temperature in a microcentrifuge. The supernatant was then filtered with a 5 μm cellulose acetate (CA) filter before being mixed 1 : 1 with 2× Binding Buffer (100 mM monosodium phosphate monohydrate pH 8.0, 30 mM Imidazole, 600 mM NaCl).

Supernatant was applied to a 5 mL HiTrap® TALON® Crude column (Cytiva) with a HiTrap® Heparin HP column (Cytiva) in series on an ÄKTA pure (Cytiva) primed with Binding Buffer (50 mM monosodium phosphate monohydrate pH 8.0, 15 mM Imidazole, 300 mM NaCl). The protein was eluted with a linear gradient of elution buffer (50 mM monosodium phosphate monohydrate pH 8.0, 200 mM Imidazole, 300 mM NaCl) and then desalted and buffer exchanged into STE buffer (10 mM Tris-HCl pH 7.5, 1 mM EDTA, 100 mM NaCl) using an Amicon Ultra-15 10 K Centrifuge Filter (Merck). The protein concentration was measured using the Qubit Protein Assay Kit and Qubit 3 Fluorometer (Thermo Fisher Scientific). The protein was then diluted in STE buffer, aliquoted, and stored at −80 °C.

**Reverse-transcriptase droplet digital PCR**. Droplet digital PCR was performed using the Bio-Rad QX200 Droplet Digital PCR system. Reactions were set up using the One-Step RT-ddPCR Advanced Kit for Probes (Bio-Rad) with primer and probe concentrations of 500 nM and 125 nM, respectively. Data were exported in CSV format and analysed using a custom Python implementation (https://github.com/mcrone/plotlydefinerain) of an online tool (http://definetherain.org.uk). The online tool uses a positive control to define positive and negative droplets using K-means clustering, with rain being determined as anything outside three standard deviations from the mean of the positive and negative clusters. It then calculates final concentration based on Eq. 1.

$$c = -ln \frac{N_{neg}}{N} / V_{droplet} \qquad (1)$$

$c$ = calculated concentration (copies/μL)
$N_{neg}$ = number of negative droplets
$N$ = total number of droplets
$V_{droplet}$ = average volume of each droplet ($0.91 \times 10^{-3}$ μL).

**Dynamic light scattering**. DLS was performed using a Zetasizer Nano (Malvern Panalytical) according to the manufacturer's instructions.

**Quantitative PCR**. qPCR experiments were designed using the combination of SAS JMP and Riffyn. Primers, probes, and their relative concentrations were those recommended by the CDC and were ordered from IDT. TaqPath 1-Step RT-qPCR Master Mix (Thermo Fisher Scientific), TaqMan Fast Virus 1-Step Master Mix (Thermo Fisher Scientific), or Luna Universal Probe One-Step RT-qPCR (NEB) were used as the relevant master mixes. qPCR reactions were otherwise set up according to the manufacturer's instructions and thermocycling settings (according to the CDC protocol). Liquid transfers were performed using an Echo 525 (Labcyte). Plates were sealed with MicroAmp Optical Adhesive Films (Thermo Fisher Scientific) and spun at 500 × g in a centrifuge. An Analytik Jena qTower³ auto was used for thermocycling and measurements were taken in the FAM channel.

**LwCas13a purification**. A plasmid expressing LwCas13 [pC013-Twinstrep-SUMO-huLwCas13a was a gift from Feng Zhang (Addgene plasmid # 90097)] was transformed into Rosetta 2 (DE3) pLysS cells (Merck Millipore). An overnight culture was inoculated into 1 L of Terrific Broth (Merck) supplemented with 35 μg/mL of Chloramphenicol (Merck) and 50 μg/mL of Kanamycin (Merck), and was grown at 37 °C, 160 r.p.m. to an OD of 0.6. The culture was then induced with 0.5 mM IPTG (Merck), cooled to 18 °C, and grown for a further 16 h. Cells were collected at 3220 × g at 4 °C and stored at −20 °C for later purification.

All protein purification steps were performed at 4 °C. The cell pellet was resuspended in lysis buffer (20 mM Tris-HCl pH 8.0, 500 mM NaCl, 1 mM dithiothreitol (DTT)) supplemented with protease inhibitors (cOmplete Ultra EDTA-free tablets, Merck) and BaseMuncher (Expedeon), and sonicated for a total of 90 s (amplitude 100% for 1 s on, 2 s off). Lysate was cleared by centrifugation for 45 min at 38,758 × g at 4 °C and the supernatant was filtered through a 5 μm CA filter.

Supernatant was applied to a 5 mL StrepTrap™ HP column (Cytiva) on an ÄKTA pure (Cytiva). The buffer of the system was changed to SUMO digest buffer (30 mM Tris-HCL pH 8, 500 mM NaCl, 1 mM DTT, 0.15% Igepal CA-630). SUMO digest buffer (5 mL) supplemented with SUMO enzyme (prepared in-house) was then loaded directly onto the column and left to incubate overnight. The cleaved protein was then eluted with 5 mL of SUMO digest buffer. The elution fraction was diluted 1 : 1 with Ion Exchange low salt buffer (20 mM HEPES pH 7, 1 mM DTT, 5% Glycerol), applied to a Hitrap® SP HP column (Cytiva), and eluted using a gradient of the ion exchange high-salt buffer (20 mM HEPES pH 7, 2000 mM NaCl, 1 mM DTT, 5% Glycerol). The eluted protein was then pooled, concentrated, and buffer exchanged into Storage buffer (50 mM Tris-HCl pH 7.5, 600 mM NaCl, 2 mM DTT, 5% Glycerol) using an Amicon Ultra-15 30 K Centrifuge Filter (Merck). The protein concentration was measured using the Qubit Protein Assay Kit and Qubit 3 Fluorometer (Thermo Fisher Scientific). The protein was then diluted, aliquoted, and stored at −80 °C.

**crRNA transcription and quantification**. DNA was ordered as ssDNA oligonucleotides from IDT and resuspended at 100 μM in Nuclease Free Duplex Buffer (IDT). Oligos contained a full-length reverse strand and a partial forward strand that contained only the T7 promoter sequence. Oligos were annealed by combining forward and reverse strands in equimolar concentrations of 50 μM and heating to 94 °C for 5 min and slow cooling (0.1 °C/s) to 25 °C in a thermocycler.

RNA was then in vitro transcribed using the TranscriptAid T7 High Yield Transcription Kit (Thermo Fisher Scientific) according to the manufacturer's instructions with a DNA template of 100 nM. Reactions were incubated for 16 h at 37 °C. DNAse I was then added and incubated for 15 min at 37 °C.

Automated purification was performed using the CyBio FeliX liquid-handling robot (Analytik Jena) using RNAClean XP beads (Beckman Coulter) according to the manufacturer's instructions.

For automated quantification, samples were loaded into a 384 PP Echo plate (Labcyte). Qubit RNA BR Dye and Qubit RNA BR Buffer (Thermo Fisher

Scientific) were premixed at a ratio of 1 : 200 and loaded into a 6-well reservoir (Labcyte). Experimental design was performed using a custom Python script and Riffyn with each sample having four technical replicates that were randomly distributed in a Greiner 384 PS Plate (Greiner Bio-One). A standard curve of 9 concentrations (0, 5, 10, 15, 20, 40, 60, 80, 100 ng/µL) was prepared using the standards provided with the Qubit RNA BR Kit (Thermo Fisher Scientific).

A volume of 9.95 µL of the mix of Qubit Dye and Qubit buffer was added to each well using an Echo 525 (Labcyte). A volume of 0.05 µL of sample was then added to each well using the Echo 525 (Labcyte) and the plate was sealed with a Polystyrene Foil Heat Seal (4titude) using a PlateLoc Thermal Microplate Sealer (Agilent). Plates were centrifuged at $500 \times g$ for 1 min before being kept in the dark for 3 min.

Plates were read using a CLARIOstar Plus (BMG Labtech) plate reader, using the following settings: excitation wavelength of 625–15 nm, dichroic of 645 nm, and emission of 665–15 nm and the Enhanced Dynamic Range (EDR) function. RNA molar concentration values were calculated, and the concentration was then normalized, RNA aliquoted and subsequently stored at $-80\,°C$.

**CRISPR-Cas13a assays with PCR amplification.** Experiments were designed and randomized using SAS JMP and Riffyn. Targets were pre-amplified using the Luna Universal One-Step RT-qPCR kit (NEB) with a primer concentration of 500 nM for 45 cycles. All concentrations are final CRISPR reaction concentrations and the final CRISPR reaction volumes were 5 µL. An Echo 525 (Labcyte) was used to transfer CRISPR Master Mix (50 nM LwCas13a, 1 U/mL murine RNAse inhibitor (NEB), 4 mM Ribonucleotide Solution Mix (NEB), 1.5 U/µl T7 RNA Polymerase (Thermo Fisher Scientific) and 1.25 ng/µL HEK293F background RNA) in Nuclease Reaction Buffer (20 mM HEPES pH 6.8, 60 mM NaCl, 9 mM $MgCl_2$) to a 384-well Small Volume LoBase Microplate (Greiner Bio-One). crRNA (25 nM) and 200 nM poly-U fluorescent probe (5′-/56-FAM/rUrUrUrUrU/3IABkFQ/-3′) were then added separately. An Echo 550 (Labcyte) was used to transfer pre-amplified products from a 384 LDV Plus Echo plate (Labcyte) to initiate the reaction, the plate was sealed, spun at $500 \times g$ for 1 min and read using a CLARIOstar Plus (BMG Labtech) plate reader with an excitation wavelength of 483-14 nm, emission of 530-30 nm, dichroic filter of 502.5 nm, and EDR enabled. Double orbital shaking of 600 r.p.m. for 30 s was performed before the first cycle. The reactions were incubated at 37 °C with readings taken every 2 min. Each reaction was normalized between a water input (background fluorescence) as 0 and an RNase I (Thermo Fisher Scientific) input (0.25 U) as 1 (RNase I cleaves all of the fluorescent probe and thus serves as a positive relative control).

**Colorimetric LAMP reactions with VLPs.** Experiments were designed and randomized using SAS JMP and Riffyn. Colorimetric LAMP reactions (NEB Warm-Start® Colorimetric LAMP 2× Master Mix) were performed with a lower final reaction volume of 5 µL. Master Mix, primers, and template were transferred to a 384-well small volume LoBase plate (Greiner Bio-One) using an Echo 525 (Labcyte). The plate was then sealed with a MicroAmp Optical Adhesive Film (Thermo Fisher Scientific) and centrifuged for 1 min at 500 g. The plate was incubated at 65 °C in a CLARIOstar Plus (BMG Labtech) plate reader and absorbance measurements were taken at 415 nm every minute for 60 min. Double orbital shaking of 600 r.p.m. for 30 s was performed before the first, sixth, and eleventh cycles.

**RNA extraction.** RNA extraction was performed using a custom Analytik Jena CyBio FeliX script (available on reasonable request) for the Analytik Jena Innu-PREP Virus DNA/RNA Kit-FX or the Promega Maxwell HT Viral TNA Kit. Samples of 200 µL were run and eluted in 50 µL of RNase Free Water.

**qPCR patient validation.** Clinical material (viral transport medium from throat/nose swabs), provided for validation by NWLP, included samples left over after clinical diagnosis as per standard practice for the validation of new assays and platforms. Patient samples were stored at room temperature for no more than 48 h after the initial analysis by NWLP before they were purified and analysed on our platform. Results (Ct values) were compared directly with those obtained by NWLP. As NWLP uses a nested PCR method, Ct values were reported as being the summation of the first and second PCR steps.

qPCR reactions were set up using the TaqPath 1-Step RT-qPCR Master Mix, CG kit, and the CDC N1 Primers according to the manufacturer's instructions and thermocycling settings (according to the CDC protocol). Final reaction volumes were 10 µL with 5 µL of extracted RNA template. Liquid transfer of the qPCR master mix was performed using an Echo 525 (Labcyte) from a 6-well reservoir (Labcyte). Extracted RNA templates were transferred using a multichannel pipette. Plates were sealed with MicroAmp Optical Adhesive Films (Thermo Fisher Scientific) and spun at $500 \times g$ in a centrifuge. An Analytik Jena qTower³ auto was used for thermocyling and measurements were taken in the FAM channel.

**CRISPR-Cas13a assays with RT-RPA amplification.** Experiments were designed and randomized using SAS JMP and Riffyn. Targets were pre-amplified using the TwistAmp Liquid Basic Kit (TwistDx) supplemented with 0.5 U/µL Murine RNase Inhibitor (NEB) and 0.08 U/µL Omniscript (Qiagen). Final reactions had a final volume of 14 µL and were set up in Echo 384 LDV Plus plates (final primer concentration of 0.45 µM and 2 µL of purified patient RNA template) and incubated at

42 °C for 30 min in a CLARIOstar Plus (BMG Labtech) plate reader with double orbital shaking of 600 r.p.m. for 30 s at 5 min. All concentrations are final CRISPR reaction concentrations and the final CRISPR reaction volumes were 5 µL. An Echo 525 (Labcyte) was used to transfer CRISPR Master Mix (50 µM LwCas13a, 1 U/µL Murine RNase inhibitor (NEB), 4 mM Ribonucleotide Solution Mix (NEB), 1.5 U/µL T7 RNA Polymerase (Thermo Fisher Scientific), and 1.25 ng/µL background RNA) in Nuclease Reaction Buffer (20 mM HEPES pH 6.8, 60 mM NaCl, 9 mM $MgCl_2$) to a 384-well Small Volume LoBase Microplate (Greiner Bio-One). crRNA (25 nM) and 200 nM poly-U fluorescent probe (5′-/56-FAM/rUrUrUrUrU/3IABkFQ/-3′) were then added separately. An Echo 550 (Labcyte) was used to transfer pre-amplified products (0.25 µL) from the 384 LDV Plus Echo plate (Labcyte) to initiate the reaction, the plate was sealed, centrifuged at $500 \times g$ for 1 min and read using a CLARIOstar Plus (BMG Labtech) plate reader with an excitation wavelength of 483-14 nm, emission of 530-30 nm, dichroic filter of 502.5 nm and EDR enabled. Double orbital shaking of 600 r.p.m. for 30 s was performed before the first cycle. The reactions were incubated at 37 °C with readings taken every 2 min. Each reaction was normalized between a water input as 0 (background fluorescence) and an RNase I (Thermo Fisher Scientific) input (0.25 U) as 1 (RNase I cleaves all of the fluorescent probe and thus serves as a positive relative control).

**Colorimetric LAMP reactions with patient samples.** Experiments were designed and randomized using SAS JMP and Riffyn. Colorimetric LAMP reactions (NEB WarmStart® Colorimetric LAMP 2× Master Mix) were performed as previously described[11] but with a lower final reaction volume of 5 µL and template of 2 µL. Master Mix, primers, and template were transferred to a 384-well Small Volume LoBase plate (Greiner Bio-One) using an Echo 525 and Echo 550 (Labcyte). The plate was then sealed with a MicroAmp Optical Adhesive Film (Thermo Fisher Scientific) and centrifuged for 1 min at 500 x g. The plate was incubated at 65 °C in a CLARIOstar Plus (BMG Labtech) plate reader and absorbance measurements were taken at 415 nm every minute for 60 min. Double orbital shaking of 600 rpm for 30 seconds was performed before the $1^{st}$, $6^{th}$, and $11^{th}$ cycles.

**Ethics statement.** Surplus clinical material was used to validate the assay as per normal practice and does not require ethical review.

**Reporting summary.** Further information on research design is available in the Nature Research Reporting Summary linked to this article.

## Data availability
Source data are provided with this paper. All other relevant data are available from the authors upon reasonable request.

## Code availability
ddPCR analysis code is available at https://github.com/mcrone/plotlydefinerain. The RNA Extraction Procotol scripts that are publically available at Zenodo at the following URL: https://doi.org/10.5281/zenodo.4021454.

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

## Acknowledgements

We thank the UK Dementia Research Institute for rapidly funding the initial workflow development and for continuing support via the UK Dementia Research Institute Care Research and Technology Centre based at Imperial College London and the University of Surrey. We also acknowledge funding from UKRI-EPSRC (EP/R014000/1, EP/S001859/1), UKRI-BBSRC (BB/M025632/1), and the National Physical Laboratory (NPL). We thank Graham Taylor, Myra McClure, and Panagiotis Pantelidis for their excellent clinical diagnostic guidance during this project and support with the surplus clinical material. We also thank Andrew Griffiths at the DRI Care Research and Technology Centre for project management support. We thank Analytik Jena and in particular Debra Conway and BMG Labtech for providing equipment and support throughout the process. We thank Professor Charles Bangham for the use of the QX200 ddPCR setup. M.A.C. thanks the Science Team at Riffyn for training and continued support. We thank Matthew Haines in the Freemont lab for critical reading of the manuscript.

## Author contributions

M.A.C., M.P., and M.C. performed experiments. M.A.C. and M.C. analysed and M.A.C., M.C., and M.S. interpreted the data. M.A.C. and M.P. created new software for this work. M.S. and P.F. supervised the work. P.R., A.A., P.F., M.S., and M.A.C. made substantial contributions to the conception and design of the work. K.J., D.S., M.A.C., M.P., M.C., M.S., and P.F. contributed to the draft of the work.

## Competing interests

M. Priestman and M. Ciechonska are co-founders of Salient Labs. All other authors declare no competing interests.
