## [Peer Review File · Nature Communications]

Reviewers' Comments:

Reviewer #1:

Remarks to the Author:

In this work, the authors simulated the detection of SARS-CoV-2 by use of a safe substitute: non-infectious synthetic virus-like particles (VLP) called MS2-SARS-CoV2 VLPs. This substitute consists of an MS2 bacteriophage capsid containing RNA from the SARS-CoV-2 N protein, which is commonly used to detect SARS-CoV-2 infection via use of the N1, N2, and N3 primer-probe sets as defined by the CDC. The authors then employed three methods to detect the presence of MS2-virus-like-particle-SARS-CoV-2: RT-qPCR, a CRISPR Cas13-based method, and colorimetric Loop-mediated isothermal Amplification (LAMP). Initially, to produce highly accurate quantification of VLP RNA content, digital droplet PCR (ddPCR) was employed. Next, RT-qPCR was used to demonstrate the utility of MS2-SARS-CoV2 VLPs as a standard for optimizing and validating diagnostic workflows and establish a lower limit of detection. CRISPR Cas13 was then used as an alternative to RT-qPCR assays, reasoning that qPCR reagents can be limited during pandemics, which necessitates the need for alternative assays. The CRISPR Cas13-based assay exhibited a comparable lower limit of detection as qPCR but was more difficult to be used in quantifying viral load. Colorimetric LAMP was able to detect MS2-SARS-CoV2 to a lower limit of 42.5 copies per reaction, significantly less sensitive than RT-qPCR or CRISPR Cas13 assays with a 17x increased lower limit of detection. Finally, the authors automate these workflows on a biofoundry and validate the automated workflows using real patient samples. In a comparison of 173 samples analyzed by North West London Pathology (NWLP) and the London Biofoundry, 169 showed agreement while the remaining 4 contentious samples were determined to be near the lowest limit of detection.

The study's main sources of novelty are the development of MS2-SARS-CoV2 VLPs, which can be used as a safe replacement for SARS-CoV2 for developing workflows and assays, and the establishment of an automated high-throughput workflow for detecting SARS-CoV2. However, some major claims will need to be supported further in order to justify publication in Nature Communications.

Major comments:

1. A major claim of this manuscript is the development of a "reagent-agnostic automated SARS-CoV-2 testing platform," specifically by using various kits for RNA extraction and RT-qPCR, which is well-supported. However, this claim is further supported by the application of two new assays, CRISPR Cas13 and LAMP, which are only sparsely used on real SARS-CoV2 patient samples. These assays should be employed on the 173 samples from NWLP and displayed alongside the results in Figure 5b. Particularly for the LAMP assay, the severely reduced sensitivity compared to qPCR or CRISPR Cas13 methods suggest this assay might have an increased false-negative rate.

2. As the main source of novelty for this manuscript is the development of MS2-SARS-CoV2 VLPs, a safe replacement for SARS-CoV2, the authors should spend more effort to directly compare SARS-CoV2 with MS2-SARS-CoV2 VLPs by demonstrating substantial equivalence in detection assays. Does the lower limit of detection established using MS2-SARS-CoV2 VLPs hold true when these assays are applied to detection of SARS-CoV2? Do these two substrates exhibit similar RNA extraction efficiencies?

Minor comment:

Rapid adoption of the automated platform used in this study by other biofoundries could contribute to the worldwide capabilities for SARS-CoV2 testing. If available, the authors are encouraged to share any scripts, worklist generation tools, or other software applications that may be rapidly adopted or used as templates by other biofoundries. As the development of an automated SARS-CoV2 platform is one of the main sources of novelty for this manuscript, providing additional tools that would allow rapid application of the author's automated workflow in other biofoundries worldwide would further justify this manuscript's publication in Nature Communications.

Reviewer #2:

Remarks to the Author:

Summary:

In the manuscript "A new role for Biofoundries in rapid prototyping, development, and validation of automated clinical diagnostic tests for SARS-CoV-2", Crone, Priestman, Ciechonska, Jensen, Sharp, Randell, Storch, and Freement describe a reagent- and automation-flexible sets of workflows for the detection of SARS-CoV-2 across three assay types. This is an important and timely contribution to the field and indeed the world in these trying times. The manuscript is well written.

Minor comments:

1) VLP expression plasmid sequence and availability.

On page 21, in the Methods "VLP preparation" subsection, the authors describe the construction and verification of the VLP expression plasmid.

However, this plasmid does not appear to have been deposited in a public physical and informatic repository (e.g. addgene, Molecular Cloud) so that readers can be certain to have physical and informatic access to this plasmid and its well-annotated DNA sequence.

The authors should deposit the physical plasmid and make its annotated DNA sequence available as an essential part of good scientific publication practices.

2) Missing software code license

On page 22, in the Methods "ddPCR" subsection, the authors reference their code repo:

<https://github.com/mcrone/plotlydefineraim>

However, this repo is missing a software license file.

Investigating the referenced website:

<http://definetherain.org.uk/>

I was led to believe that the software (a naïve initial misunderstanding on my part) would be under an MIT license, and was then pointed to this distinct repository:

<https://github.com/jacobhurst/definetherain/blob/master/LICENSE>

which does in fact include a license file (MIT).

The authors should state the software license (e.g. MIT) in the manuscript, and include the corresponding license file in their repository.

Reviewer #3:

Remarks to the Author:

The authors describe multiple techniques for SARS-CoV-2 viral detection and how these can be scaled for high throughput. This paper is timely and compares the three main viral detection methods. The authors validate multiple brands of reagents important as availability varies worldwide. Finally, the authors test methods using both a VLP and patient samples. I commend the authors for using a VLP and quickly evaluating multiple methods and using different commercially available reagents.

I think this paper is timely and a great comparison of many techniques. It would benefit from a

table comparing all the data and techniques. The info that I personally would find helpful: assay, commercial reagents (if applicable), LOD, time to result, HT techniques, advantages and disadvantages. This is the first paper to compare all three viral tests and with a little refinement I think it will garner many citations as more labs begin to publish their results.

Throughout the paper there is a lot of redundancy and the manuscript would benefit from a paring down. There are also grammar errors and sentences with redundant words.

There are some minor changes and edits that would clarify the work.

1. I would more clearly define what "full synthetic biology stack available at the London Biofoundry" actually is. I had to google to understand what it was. I would define it once clearly in the introduction instead of leaving this until the discussion. .

2. The introduction is quite long

3. In the introduction update statistics on cases that are now dated.

4. There have already been multiple CRISPR-Cas and LAMP assay submitted to BioRxiv. Please address in the intro and/or page 9-11 either through citations or a quick comparison. The authors do this well in the discussion, but it would be clearer if this was highlighted in the introduction.

5. Was the full N gene OR a segment complementary to the CDC primers of the N gene used? In the introduction paragraph 5 it appears it was part of the gene while in the results it seems to be stated as the full N gene. I would clarify this.

6. Clearly state at first mention that the primer-probes used throughout for RT-qPCR are CDC. Sometimes this is stated and other times not so it becomes confusing.

7. Figure 1c – the error bars are not clear. Even when the figure is blown up. Recommend changing colors or bolding.

8. For the VLP validation by RT-qPCR. Was this automated? If not clearly state. If so, I would highlight more.

9. For CRISPR-Cas work use CRISPR-Cas13a as there are other Cas's. guide RNA is not gRNA but referred to as crRNA-guided or just crRNA. Clarify your method requires a plate reader.

10. How was the CRISPR-Cas work normalized? For Figs 3b,4c fluorescence is usually reported as fluorescence a.u.

11. Page 10 - If samples need to be loaded manually then 1000 samples in 12 hours seems high. Please review and clarify if this is really being completed at scale in hospitals.

12. Page 15 and in methods. Report master-mixes in ul

13. Page 16 needs a bit of work. First were there any freeze thaws or potential changes to samples between the lab test and the NWLP tests? Paragraph 2. Are these completely different patients' samples? Why? State numbers clearly – based on Figure 5c the Promega test was far fewer samples. If using a range of Ct's to define low, medium and high – state the exact range both in text and in figure.

We thank the Reviewers for their constructive and thoughtful comments which we address below.

Reviewer 1	1. A major claim of this manuscript is the development of a “reagent-agnostic automated SARS-CoV-2 testing platform,” specifically by using various kits for RNA extraction and RT-qPCR, which is well-supported. However, this claim is further supported by the application of two new assays, CRISPR Cas13 and LAMP, which are only sparsely used on real SARS-CoV2 patient samples. These assays should be employed on the 173 samples from NWLP and displayed alongside the results in Figure 5b.
	We thank the reviewer for this suggestion. The primary use of patient samples reported in the manuscript was for the validation of our own and the Imperial Molecular Diagnostics Unit (MDU) RNA extraction and qPCR workflows. The leftover RNA obtained from these extractions was then used for an important Public Health England sequencing project. Therefore, we were not able to use these same samples, when we performed the CRISPR and LAMP testing later on. Adding to the precious nature of patient samples in the current circumstances, the majority of traditional patient testing at our NPLW’s diagnostic labs has been using the Roche cobas platform which does not allow for leftover RNA collection at the end of the workflow. This results in a lack of patient sample RNA that can be used to validate new workflows. Our intention for the CRISPR and LAMP demonstrations was to show that they are suitable alternatives for population testing given that they have lower sensitivity. CRISPR is very dependent on the preamplification technique, PCR preamplification shows the same sensitivity as for RT-qPCR as seen in Figure 3 and 4. However, RPA and LAMP preamplification (isothermal) have lower sensitivity as seen in testing our new samples and as seen in the IDT Sherlock Cas13a CDC EUA testing kit documentation. We have processed and analysed an additional 24 patient samples (11 RT-qPCR positives) to further characterise our initial proof of concept work for LAMP and CRISPR. We show Ct values for RT-qPCR and the LAMP and CRISPR results for 3 amplification replicates per patient sample. We also show the number of copies per reaction after fitting the RT-qPCR data to a standard curve with our VLPs to estimate the lower limit of detection for LAMP and RPA-CRISPR.
	Particularly for the LAMP assay, the severely reduced sensitivity compared to qPCR or CRISPR Cas13 methods suggest this assay might have an increased false-negative rate.
	We thank the reviewer for this very good point - the previously presented lowest copy number detected for the LAMP assay was the lowest copy number attempted in this experiment. To address this important question, we have repeated the LAMP experiment including lower VLP copy numbers, to measure the lower limit of detection. These results have now been added to the manuscript (Figure 3d).
	2. As the main source of novelty for this manuscript is the development of MS2-SARS-CoV2 VLPs, a safe replacement for SARS-CoV2, the authors should spend more effort to directly compare SARS-CoV2 with MS2-SARS-CoV2 VLPs by demonstrating substantial equivalence in detection assays. Does the lower limit of detection established using MS2-SARS-CoV2 VLPs hold true when these assays are

	applied to detection of SARS-CoV2? Do these two substrates exhibit similar RNA extraction efficiencies?
	We thank the reviewer for this suggestion. In order to compare the extraction efficiency of the MS2 VLPs and SARS-CoV-2 we would need to culture live SARS-CoV-2 in tissue culture and purify and quantify the virus. Since SARS-CoV-2 is a Hazard Group 3 organism this would require Biosafety Level 3 facilities, which we do not have access to. Once the virus has been produced and purified it would then need to be quantified accurately. To replicate our VLP quantification strategy, we would have to quantify the virus performing RT-ddPCR in a Biosafety Level 3 facility. Given these substantial efforts required we would rather emphasize that the goal of this work was to rapidly develop and compare automated SARS-CoV-2 testing methods. The development of most diagnostic workflows relies on the use of naked RNA that is spiked into biological matrices and we feel that this is a poor model for the manipulation of viral particles and introduces new risks to assay reliability given the properties of unprotected RNA. Therefore, we developed the MS2-SARS-CoV-2 VLPs to help create and optimise our workflows and assays at Biosafety Level 1 before deploying them for use with patient samples. We show that the workflows developed and optimized using the MS2-SARS-CoV-2 VLPs directly translate and provide reliable analysis of patient samples, when compared to current diagnostic workflows. We argue therefore, that showing our workflows developed and informed by their performance on SARS-Cov-2 VLPs are then also working very well in clinical settings on patient samples (as shown in Figure 5) is a strong validation for the use and application of MS2-SARS-CoV-2 VLPs for rapidly developing reliable new diagnostic workflows.
	Minor comment: Rapid adoption of the automated platform used in this study by other biofoundries could contribute to the worldwide capabilities for SARS-CoV2 testing. If available, the authors are encouraged to share any scripts, worklist generation tools, or other software applications that may be rapidly adopted or used as templates by other biofoundries. As the development of an automated SARS-CoV2 platform is one of the main sources of novelty for this manuscript, providing additional tools that would allow rapid application of the author's automated workflow in other biofoundries worldwide would further justify this manuscript's publication in Nature Communications.
	We will make updated versions of the protocols and scripts available on https://github.com/LondonBiofoundry/ and they are already available upon request. For instance, we already shared materials with three Biofoundries through the Global Biofoundry Alliance.
Reviewer 2	VLP expression plasmid sequence and availability. On page 21, in the Methods "VLP preparation" subsection, the authors describe the construction and verification of the VLP expression plasmid. However, this plasmid does not appear to have been deposited in a public physical and informatic repository (e.g. addgene, Molecular Cloud) so that readers can be certain to have physical and informatic access to this plasmid and its well-annotated DNA sequence. The authors should deposit the physical plasmid and make its annotated DNA sequence available as an essential part of good scientific publication practices.

	The generic expression vector with Type IIs restriction sites (for insertion of any RNA sequence) is already available on addgene (https://www.addgene.org/128233/) and the plasmid expressing the MS2-SARS-CoV-2 N gene VLP used in this work (https://www.addgene.org/155039/) and another encoding for the MS2-SARS-CoV-2 E gene VLP (https://www.addgene.org/155040/) have been submitted to addgene and are awaiting QC.
	2) Missing software code license On page 22, in the Methods “ddPCR” subsection, the authors reference their code repo: https://github.com/mcrones/plotlydefineraim However, this repo is missing a software license file. Investigating the referenced website: http://definetherain.org.uk/ I was led to believe that the software (a naïve initial misunderstanding on my part) would be under an MIT license, and was then pointed to this distinct repository: https://github.com/jacobhurst/definetherain/blob/master/LICENSE which does in fact include a license file (MIT). The authors should state the software license (e.g. MIT) in the manuscript, and include the corresponding license file in their repository.
	Thank you for carefully reviewing the scripts. This is an important point and the https://github.com/mcrones/plotlydefineraim repository now has a license file attached and further instructions for use have been added.
Reviewer 3	1. I would more clearly define what “full synthetic biology stack available at the London Biofoundry” actually is. I had to google to understand what it was. I would define it once clearly in the introduction instead of leaving this until the discussion.
	Thank you for your suggestion, we agree and defined the full synthetic biology stack in the introduction.
	2. The introduction is quite long
	We agree and condensed the introduction.
	3. In the introduction update statistics on cases that are now dated.
	We agree and included updated statistics to reflect the case numbers on the 8th of July.
	4. There have already been multiple CRISPR-Cas and LAMP assay submitted to BioRxiv. Please address in the intro and/or page 9-11 either through citations or a quick comparison. The authors do this well in the discussion, but it would be clearer if this was highlighted in the introduction.
	We agree and added citations for the relevant BioRxiv articles in the introduction.
	5. Was the full N gene OR a segment complementary to the CDC primers of the N gene used? In the introduction paragraph 5 it appears it was part of the gene while in the results it seems to be stated as the full N gene. I would clarify this.
	Thank you for this suggestion, we have clarified that the entire N gene was encoded in the VLP standard.
	6. Clearly state at first mention that the primer-probes used throughout for RT-qPCR are CDC. Sometimes this is stated and other times not so it becomes confusing.
	Thank you for this suggestion – we have clarified this throughout the manuscript.

	7. Figure 1c – the error bars are not clear. Even when the figure is blown up. Recommend changing colors or bolding.
	Thank you for this suggestion – we have increased the size of the plot to make it easier to see the error bars.
	8. For the VLP validation by RT-qPCR. Was this automated? If not clearly state. If so, I would highlight more.
	Thank you for this suggestion – it was automated and we added to the sentence for clarity.
	9. For CRISPR-Cas work use CRISPR-Cas13a as there are other Cas's. guide RNA is not gRNA but referred to as crRNA-guided or just crRNA. Clarify your method requires a plate reader.
	We agree - for clarity CRISPR-Cas has been changed to CRISPR-Cas13a and gRNA has been changed to crRNA. We clarified in the methods that a plate reader is required.
	10. How was the CRISPR-Cas work normalized? For Figs 3b,4c fluorescence is usually reported as fluorescence a.u.
	Thank you for requesting this clarification – Measurements were taken as arbitrary units specific to plate reader type and gain settings. Therefore, we use RNase I (ThermoFisher Scientific) to create a run control by cleaving all of the fluorescent reporter in a positive fluorescence control well. Water is then added to a CRISPR reaction without target RNA to obtain the background increase in fluorescence in the absence of CRISPR-Cas13a activation. Sample results are then then normalised against these two controls with the RNase I control assumed as 1 and the Water CRISPR reaction assumed 0. We have expanded this explanation in the methods for clarity.
	11. Page 10 - If samples need to be loaded manually then 1000 samples in 12 hours seems high. Please review and clarify if this is really being completed at scale in hospitals.
	Thanks for raising this point- the throughput of the complete testing framework does depend on the upstream capability at a given testing site. In the context of the NWLP hospital lab where one of our platforms is in use the upstream manual sample handling is actually not too difficult as samples are already in deactivation buffer and therefore bypassing the slow BSL3 working environment. So 1000 samples in 12 hours can be achieved. In fact this hospital site has now committed to expanding the platform for a 3000 samples/24h testing capacity.
	12. Page 15 and in methods. Report master-mixes in ul
	This has been corrected.
	13. Page 16 needs a bit of work. First were there any freeze thaws or potential changes to samples between the lab test and the NWLP tests? Paragraph 2. Are these completely different patients' samples? Why? State numbers clearly – based on Figure 5c the Promega test was far fewer samples. If using a range of Ct's to define low, medium and high – state the exact range both in text and in figure.
	Thanks for requesting this useful clarification – Tested samples were stored for no more than 48 hours at room temperature before being analysed on our platform (after initial analysis by NWLP). We have added this to the manuscript for clarity. When validating the initial workflow, a larger number of patient samples has been used because the entire workflow needed to be validated from start to finish for UKAS accreditation. However, once this initial validation has taken place, a smaller sample pool can be used for “verification” if only one part of the process is changed (i.e. the extraction kit). The number of patient samples has been added to the manuscript for clarity.

	The samples used for LAMP and CRISPR are different from the samples used for the initial qPCR workflow validation. The extracted RNA from the initial sample set had to be used for an important Public Health England sequencing project and so we had to use a different set of samples available at the time. To address the valid request of comparing the test results across all detection methods based on the same set of extracted RNA samples, we repeated the work with 24 patient samples (11 positive on RT-qPCR). The number of copies in each reaction has been summarised in a new supplementary table using the RT-qPCR results and standard curve generated using the VLPs. We have removed the labelling of low, medium and high Ct values to avoid confusion.
--	--

Reviewers' Comments:

Reviewer #1:

Remarks to the Author:

The authors have addressed all of my critiques satisfactorily. I would recommend the acceptance of this manuscript for publication. The work is timely and of interest to the broad research community and general public.